

# Static and dynamical signatures of Dzyaloshinskii-Moriya interactions in the Heisenberg model on the kagome lattice

Francesco Ferrari[1,2⋆], Sen Niu[3⋆], Juraj Hasik[4],
Yasir Iqbal[2], Didier Poilblanc[3] and Federico Becca[5]

**1** Institut für Theoretische Physik, Goethe Universität Frankfurt,
Max-von-Laue-Straße 1, D-60438 Frankfurt am Main, Germany
**2** Department of Physics and Quantum Centers in Diamond and Emerging Materials
(QuCenDiEM) group, Indian Institute of Technology Madras, Chennai 600036, India
**3** Laboratoire de Physique Théorique, Université de Toulouse, CNRS, UPS, France
**4** Institute for Theoretical Physics, University of Amsterdam,
Science Park 904, 1098 XH Amsterdam, The Netherlands
**5** Dipartimento di Fisica, Università di Trieste, Strada Costiera 11, I-34151 Trieste, Italy

⋆ These authors contributed equally

## Abstract

Motivated by recent experiments on $Cs_2Cu_3SnF_{12}$ and $YCu_3(OH)_6Cl_3$, we consider the $S = 1/2$ Heisenberg model on the kagome lattice with nearest-neighbor super-exchange $J$ and (out-of-plane) Dzyaloshinskii-Moriya interaction $J_D$, which favors (in-plane) $Q = (0,0)$ magnetic order. By using both variational Monte Carlo and tensor-network approaches, we show that the ground state develops a finite magnetization for $J_D/J \gtrsim 0.03{-}0.04$; instead, for smaller values of the Dzyaloshinskii-Moriya interaction, the ground state has no magnetic order and, according to the fermionic wave function, develops a gap in the spinon spectrum, which vanishes for $J_D \to 0$. The small value of $J_D/J$ for the onset of magnetic order is particularly relevant for the interpretation of low-temperature behaviors of kagome antiferromagnets, including $ZnCu_3(OH)_6Cl_2$. For this reason, we assess the spin dynamical structure factor and the corresponding low-energy spectrum, by using the variational Monte Carlo technique. The existence of a continuum of excitations above the magnon modes is observed within the magnetically ordered phase, with a broad peak above the lowest-energy magnons, similarly to what has been detected by inelastic neutron scattering on $Cs_2Cu_3SnF_{12}$.



# 1 Introduction

The Heisenberg model represents the simplest and most idealized way to describe the interaction among localized magnetic moments in a solid. It has been pivotal to explain conventional magnetic phase transitions, but also a wide range of unconventional phenomena, including the existence of topological phases, e.g., in two-dimensional systems with O(2) spin symmetry at finite temperature [1], and one-dimensional models with integer spin values at zero temperature [2]. Over the last 20 years, the interest shifted towards the investigation of frustrated spin systems, aiming to clarify the possibility of realizing quantum spin-liquid phases, which are characterized by long-range entanglement and sustain fractional excitations and (for gapped phases) topological order [3, 4]. Due to considerable recent developments of numerical methods, it is now possible to study the effect of different relevant perturbations on top of the pure Heisenberg interaction, reaching a high level of accuracy. For example, multi-spin interactions have been considered [5–7]. In this regard, chiral terms have also been investigated, to assess the possibility to stabilize bosonic analogues of the fractional quantum Hall states [8]. In addition, models with spatially-anisotropic exchange interactions (generated by a spin-orbit entanglement) have been explored [9–11], also motivated by Kitaev's seminal work, which defined an exactly-solvable model with bond-dependent Ising-like interactions that hosts a spin-liquid ground state [12].

A particularly relevant interaction for several magnetic materials is the Dzyaloshinskii-Moriya (DM) term [13, 14], an anti-symmetric exchange coupling, which originates from spin-orbit effects. A non-vanishing DM interaction, which explicitly breaks the SU(2) spin symmetry, can only exist in structural geometries without bond-inversion symmetry. In this regard, the effect of the DM interaction on top of the Heisenberg model in the kagome lattice has been recently investigated, because of its relevance for a number of $S = 1/2$ materials, e.g., $ZnCu_3(OH)_6Cl_2$, $YCu_3(OH)_6Cl_3$, and $Cs_2Cu_3SnF_{12}$. The first one, commonly known as *Herbertsmithite*, represents a particularly important compound, since its low-temperature behavior is compatible with the existence of a gapless (or weakly gapped) spin liquid [15–18]; here, in addition to the nearest-neighbor coupling $J \approx 180K$ [19], a small out-of-plane DM interaction, $J_D/J \approx 0.04-0.08$ may be present [20,21]. The second and third compounds are instead magnetically ordered, with a pitch vector $\mathbf{Q} = (0, 0)$ in the kagome planes [22,23]; the

magnetic order is ascribed to the presence of a relatively large (out-of-plane) DM interaction, i.e., $J_D/J \approx 0.18$ for $Cs_2Cu_3SnF_{12}$ and $J_D/J \approx 0.25$ for $YCu_3(OH)_6Cl_3$.

From the theory side, the effect of the DM interaction in the Heisenberg model on the kagome lattice has been investigated in several works. An early exact diagonalization study [24] suggested that the magnetically ordered phase is stabilized for $J_D/J \gtrsim 0.1$. A similar outcome has also been obtained within functional renormalization-group approach [25]. By contrast, recent tensor-network (TN) calculations [26] have found that magnetic order sets in for $J_D/J \gtrsim 0.012(2)$, while a gapless spin liquid exists for smaller values of the DM interaction (as in the nearest-neighbor Heisenberg model [27–29]). Mean-field studies, based upon Schwinger-boson [30,31] and Abrikosov-fermion [32,33] approaches, have been employed to assess the magnetically disordered phase, including the possible existence of a $\mathbb{Z}_2$ chiral spin liquid [34]. In addition, the spectral properties have been investigated by exact diagonalization (at finite temperature) [35] and density-matrix renormalization group [36], to evaluate the evolution of the low-energy modes towards the onset of magnetic order.

In this work, we first revisit the phase diagram of the nearest-neighbor Heisenberg model on the kagome lattice in presence of an out-of-plane DM interaction. We employ both a variational Monte Carlo (VMC) technique, based upon Gutzwiller-projected fermionic states [37], and TN algorithms, based on infinite projected-entangled pair states (iPEPS) [38] and infinite projected-entangled simplex states (iPESS) [39,40]. A consistent estimation of the transition point between the disordered phase and the $\mathbf{Q} = (0,0)$ ordered phase is obtained by these approaches, i.e., $J_D/J = 0.030(5)$ within VMC and $0.040(5)$ within iPEPS and iPESS. The small, but significant, discrepancy between our TN estimation of the critical point and the one obtained in Ref. [26] can be ascribed to different optimization and extrapolation schemes. In this work, we employ the algorithmic differentiation [41] to optimize the tensors variationally and the finite-correlation length scaling [42,43] to perform the thermodynamic extrapolations. In addition to ground-state properties, we assess the dynamical structure factor of the system by employing the dynamical VMC method, proposed in Ref. [44] and recently applied to a variety of frustrated Heisenberg models [45–51]. Here, we extend this approach to compute both in-plane and out-of-plane spin-spin correlations in the model with DM terms, which explicitly break the SU(2) spin symmetry. The low-energy spectrum shows an extended continuum with a broad peak just above the (damped) gapped magnons, which is particularly visible around the mid-point of the edges of the extended Brillouin zone. This feature resembles what has been detected in inelastic neutron scattering experiments on $Cs_2Cu_3SnF_{12}$ [23].

## 2 Model and Methods

The spin Hamiltonian is defined by

$$\mathcal{H} = J \sum_{\langle i,j \rangle} \mathbf{S}_i \cdot \mathbf{S}_j + J_D \sum_{\overrightarrow{\langle i,j \rangle}} \left( S_i^x S_j^y - S_i^y S_j^x \right), \tag{1}$$

where $\mathbf{S}_i = (S_i^x, S_i^y, S_i^z)$ is the $S = 1/2$ spin operator on site $i$ and $\langle \dots \rangle$ indicates nearest-neighboring sites. The super-exchange coupling $J$ is invariant under $i \leftrightarrow j$, while the DM term $J_D$ changes sign when exchanging $i$ and $j$, thus an orientation of the bonds needs to be specified in order to fully determine the Hamiltonian. The orientations, denoted by $\overrightarrow{\langle i,j \rangle}$, are represented by the arrows sketched in Fig. 1.

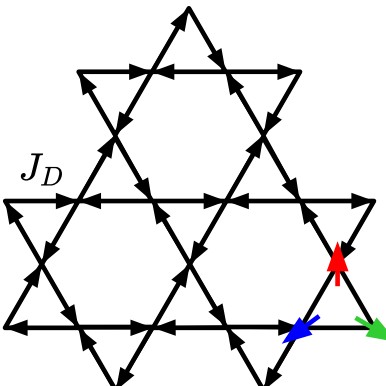

Figure 1: Definition of the bond orientations $\overrightarrow{\langle i, j \rangle}$ that determine the sign of the out-of-plane DM interactions in the Hamiltonian of Eq. (1). The orientation of the three spins in the unit cell for the $\mathbf{Q} = (0, 0)$ magnetic order is also shown.

## 2.1 Tensor-network states

TN calculations are based upon iPEPS [38] and iPESS [39,40], which are constructed from an effective square lattice and a few tensors that contain the variational parameters. In particular, the iPEPS *Ansatz* approximates the ground states by a single rank-5 tensor $a$ placed at every down-pointing triangle of kagome lattice, see Fig. 2(a). The tensor $a^s_{uldr}$ has a physical index $s$ of dimension $d_p = 2^3$, which runs over all spin states of the down-pointing triangle, and four auxiliary indices $u$, $l$, $d$, and $r$, with bond dimension $D$ corresponding to the directions of the square lattice. Then, the total number of variational parameters of this iPEPS is $d_p \times D^4$. Instead, the iPESS *Ansatz* restricts the form of on-site tensor $a$, defining it by a contraction of five rank-3 tensors, see Fig. 2(b). Two *trivalent* tensors $t_u$ and $t_d$, each with three auxiliary indices of bond dimension $D$, encode states of virtual degrees of freedom associated to up- and down-pointing triangles. Three *bond* tensors $b_1$, $b_2$, and $b_3$ host the physical spin-1/2 degrees of freedom and connect these trivalent tensors. Each bond tensor has two auxiliary indices of bond dimension $D$ and a physical index of dimension 2. Therefore, the number of variational parameters in the iPESS *Ansatz* is $2 \times D^3 + 3 \times 2D^2$. The iPESS is constructed to treat bonds of kagome lattice faithfully, at the expense of variational freedom, whereas iPEPS favours bonds within up-pointing triangles. With growing $D$ these *Ansätze* are expected to reconcile. The expectation values of the Hamiltonian and local observables (e.g., $m^2$, the square of the magnetization) are evaluated using the corner-transfer matrix (CTM) method [52]. The CTM approximates the contraction of infinite networks by using a set of environment tensors $\{C, T\}$ with characteristic size $\chi$, dubbed environment bond dimension. The elements of these tensors are highly non-linear functions of the original tensor $a$. The variational energy is minimized by optimizing the elements of $a$ (for iPEPS) or tensors $t_u$, $t_d$, $b_1$, $b_2$, and $b_3$ (for iPESS), by employing the gradient descent method. The gradients are evaluated by automatic differentiation. The implementation of these *Ansätze* with CTM method and their optimization is provided by the *peps-torch* library [53]. In general, for fixed bond dimension $D$, iPEPS has lower variational energy and smaller magnetization than iPESS (since the iPEPS state has more variational parameters than the iPESS one). Quantitative comparisons between iPEPS and iPESS are reported in the Appendix A.

All the tensor-network calculations are performed with finite values of the CTM environment dimension $\chi$ and the bond dimension $D$. The thermodynamic limit is then reached by taking $\chi \to \infty$ for each $D$ and then $D \to \infty$. To achieve the limit of $D \to \infty$, we use the finite-correlation length scaling [42,43], which gives more accurate results than the straight-

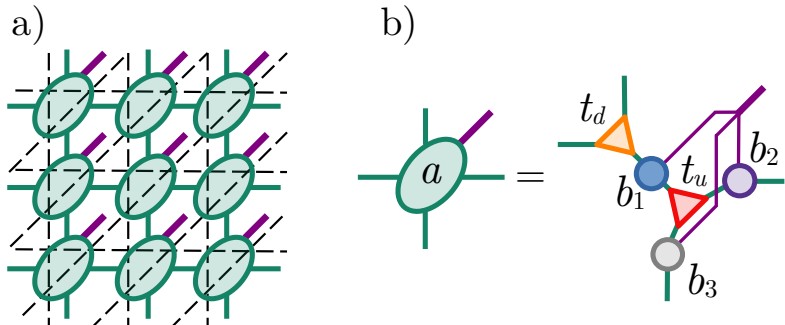

Figure 2: TN *Ansätze* for the kagome lattice. (a) The iPEPS state is defined by a single rank-5 on-site tensor $a$ associated to every down-pointing triangle of kagome lattice. The physical index of $a$ (violet) with dimension $d_p = 2^3$ runs over all states of the three spin-1/2 degrees of freedom on vertices of down-pointing triangles. (b) The iPESS state is defined by the on-site tensor $a$, which is obtained by a contraction of five rank-3 tensors: Two trivalent tensors $t_u$ and $t_d$ associated to up- and down-pointing triangles, respectively, and three bond tensors $b_1$, $b_2$, and $b_3$, each one with single physical index (violet) representing one of the spin-1/2 degrees of freedom on vertices of down-pointing triangles. All auxiliary indices (green) of iPEPS and iPESS have bond dimension $D$.

forward $1/D$ extrapolation. Here, the correlation length $\xi$ of an infinite-size TN state plays the role of infrared cut-off, analogously to the linear lattice size of numerical methods based on finite-size calculations. An important aspect of our scaling analysis is that the same thermodynamic value of $m^2$ is imposed for both iPEPS and iPESS data, since they are expected to converge to the same (exact) ground state. For the magnetic phase with relatively large magnetization (i.e., for large values of $J_D/J > 0.1$), we consider the fitting functions:

$$m^2_{\text{iPEPS}}(\xi) = m^2(\infty) + a/\xi, \tag{2}$$

$$m^2_{\text{iPESS}}(\xi) = m^2(\infty) + b/\xi. \tag{3}$$

By contrast, within the spin-liquid and weakly-ordered regimes (i.e., for small values of $J_D/J \leq 0.1$), an empirical fit turns out to be more suitable:

$$m^2_{\text{iPEPS}}(\xi) = m^2(\infty) + a/\xi^b, \tag{4}$$

$$m^2_{\text{iPESS}}(\xi) = m^2(\infty) + c/\xi^d. \tag{5}$$

The smoothness of the scaling in both spin-liquid and magnetically ordered phases is demonstrated in Fig. 3 for a few values of $J_D/J$. The value of the thermodynamic magnetization is obtained by performing different extrapolations using different sets of points. The final $m^2$ value corresponds to the best fit (i.e., the fit with smallest fitting error) and the error bar is determined by combining the results of the different extrapolations. We emphasize that the variational optimization provides data with higher quality then the simple-update method used in Ref. [26] and is crucial for extrapolations, see Appendix A for a detailed discussion.

## 2.2 Gutzwiller-projected wave functions

VMC calculations are performed by using Gutzwiller-projected wave functions [54–56], constructed from the Abrikosov-fermion representation of the spin operators [57–59],

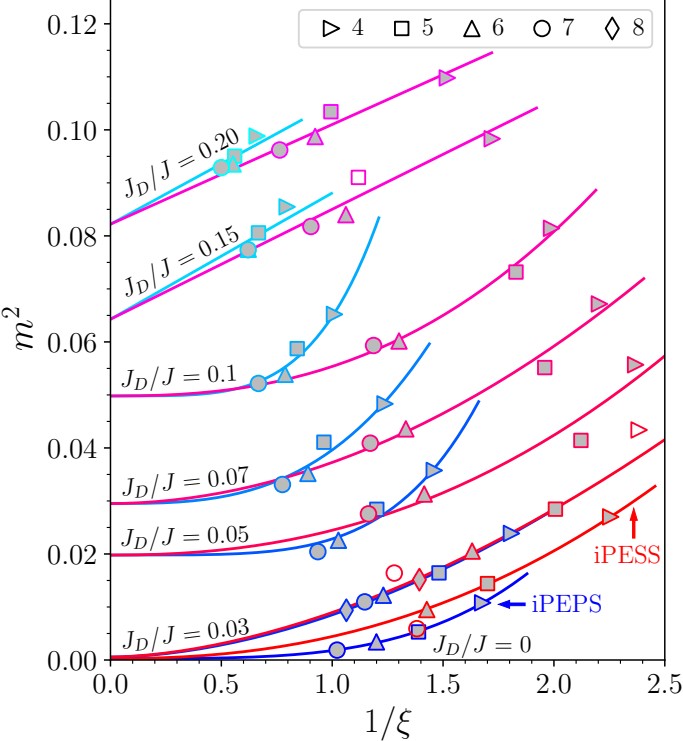

Figure 3: Finite-correlation length scaling of the antiferromagnetic order parameter (squared) $m^2$, for both iPEPS and iPESS *Ansätze* with bond dimension $D$ ranging from 4 to 8. The power-law fittings of Eqs. (2), (3), (4), and (5) are employed. The empty markers are excluded in the fits shown here, but contribute to the determination of the error bar on the extrapolated results.

$\mathbf{S}_i = \frac{1}{2}\sum_{\alpha,\beta} c^\dagger_{i,\alpha} \boldsymbol{\sigma}_{\alpha,\beta} c_{i,\beta}$, and the definition of an auxiliary Hamiltonian containing hoppings and a Zeeman field:

$$\mathcal{H}_0 = \sum_{\langle i,j\rangle,\alpha} \chi^\alpha_{ij} c^\dagger_{i,\alpha} c_{j,\alpha} + h\sum_i \mathbf{M}_i \cdot \mathbf{S}_i . \tag{6}$$

The first term is a nearest-neighbor hopping, including both "singlet" ($\chi^\uparrow_{ij} = \chi^\downarrow_{ij}$) and "triplet" ($\chi^\uparrow_{ij} = -\chi^\downarrow_{ij}$) complex-valued amplitudes [33]. In particular, we find that the optimal variational *Ansatz* contains a real singlet hopping and a purely imaginary triplet hopping [33], which reduces to the $U(1)$ Dirac state [60,61] when restricted to the singlet part. The second term (for $h \neq 0$) induces magnetic order in the $XY$ plane, with the periodicity determined by the unit vector $\mathbf{M}_i = [\cos(\mathbf{Q}\cdot\mathbf{R}_i + \phi_i), \sin(\mathbf{Q}\cdot\mathbf{R}_i + \phi_i), 0]$ (where $\mathbf{Q}$ is the pitch vector, $\mathbf{R}_i$ is the coordinate of the unit cell of site $i$, and $\phi_i$ is a sublattice-dependent angle). In the following, we consider the $\mathbf{Q} = (0,0)$ case, with $\phi_i$ giving 120° order in all triangles (as sketched in the inset of Fig. 1), which is suitable for the out-of-plane DM interaction. The full variational wave function $|\Psi_0\rangle$ is built from the ground state $|\Phi_0\rangle$ of the Hamiltonian (6), applying the Gutzwiller projector, which enforces single fermionic occupations, $\mathcal{P}_G = \prod_i (n_{i,\uparrow} - n_{i,\downarrow})^2$, where $n_{i,\alpha} = c^\dagger_{i,\alpha} c_{i,\alpha}$. In addition, a projector onto the subspace with $S^z = \sum_i S^z_i = 0$ and a spin-spin Jastrow factor $\mathcal{J} = \exp\left(1/2 \sum_{i,j} v_{i,j} S^z_i S^z_j\right)$ (where $v_{i,j}$ are variational parameters) are included:

$$|\Psi_0\rangle = \mathcal{J}\mathcal{P}_{S_z=0}\mathcal{P}_G|\Phi_0\rangle . \tag{7}$$

Calculations are done on $N = 3 \times L \times L$ clusters and extrapolations to the thermodynamic

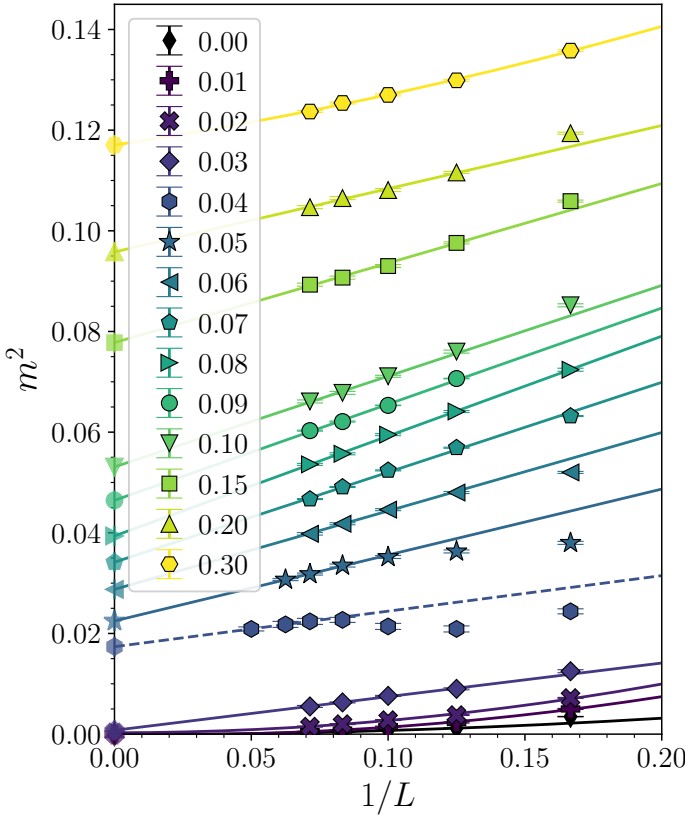

Figure 4: Finite-size scaling of the antiferromagnetic order parameter (squared) $m^2$ as computed by VMC. Different colors/markers indicate different values of $J_D/J$, as reported in the box.

limit are performed using standard finite-size scaling analysis [62, 63]. In Fig. 4, we show the extrapolation of the order parameter $m^2$ as a function of $L$, for several values of the ratio $J_D/J$. For $J_D/J < 0.03$, the magnetization curves are fitted with a $1/L^2$ behavior.

Within the present variational approach, we can also define a set of excitations to approximate the low-energy spectrum of the system and compute the dynamical spin-spin correlations [44]. In fact, approximate excited states are constructed by linear superpositions of particle-hole (spinon) excitations, whose coefficients are determined by the Rayleigh-Ritz variational principle. Although the recipe to define two-spinon excitations with $S^z = 0$ has been discussed in details in recent works [46, 47, 50] on frustrated spin models (with SU(2) spin symmetry), here we briefly outline the general construction for the case of a Bravais lattice with a basis, which is suitable for the kagome lattice under investigation (technical details can be found in Ref. [48]). For this purpose, we adopt an explicit notation in which sites are denoted by a Bravais vector $\mathbf{R}$ and a sublattice index $a$, such that $c_{i,\beta} \to c_{R,a,\beta}$. A (non-orthogonal) set of particle-hole spinon excitations with momentum $\mathbf{q}$ is defined by the states

$$|q;R,a;b\rangle = \mathcal{J}\mathcal{P}_G\mathcal{P}_{S^z=0} \sum_{R'} e^{i\mathbf{q}\cdot\mathbf{R}'} \left( c^\dagger_{R+R',a,\uparrow} c_{R',b,\uparrow} - c^\dagger_{R+R',a,\downarrow} c_{R',b,\downarrow} \right) |\Phi_0\rangle. \tag{8}$$

Then, the approximate excited states for the spin model are taken as linear combinations of $\{|q;R,a;b\rangle\}$ states

$$|\Psi^q_n\rangle = \sum_R \sum_{a,b} A^{n,q}_{R,a;b} |q;R,a;b\rangle, \tag{9}$$

where $n$ is an integer index labelling the variational excitations with momentum $\mathbf{q}$. The coef-

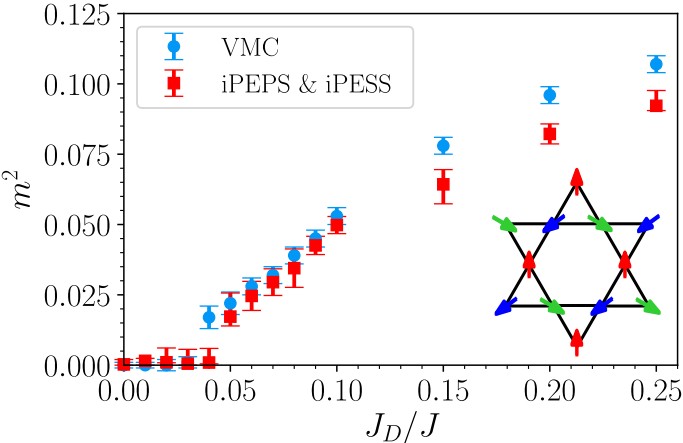

Figure 5: Antiferromagnetic order parameter (squared) at $\mathbf{Q} = (0,0)$ as a function of $J_D/J$. Both VMC and TN results are reported, as extrapolated in the thermodynamic limit (which is obtained by standard finite-size extrapolation $L \to \infty$ for VMC and finite-correlation-length scaling $\xi \to \infty$ for TN, combining iPEPS and iPESS results). The inset shows the $\mathbf{Q} = (0,0)$ coplanar order induced by the presence of the out-of-plane DM interaction.

ficients of the expansion, $A_{R,a;b}^{n,q}$, are obtained by solving the generalized eigenvalue problem

$$\sum_{R',a',b'} H_{R,a;b|R',a';b'}^q A_{R',a';b'}^{n,q} = E_n^q \sum_{R',a',b'} O_{R,a;b|R',a';b'}^q A_{R',a';b'}^{n,q}, \tag{10}$$

where $E_n^q$ are the energies of the excitations $|\Psi_n^q\rangle$, and

$$H_{R,a;b|R',a';b'}^q = \langle q;R,a;b|\mathcal{H}|q;R',a';b'\rangle, \tag{11}$$

$$O_{R,a;b|R',a';b'}^q = \langle q;R,a;b|q;R',a';b'\rangle, \tag{12}$$

are the Hamiltonian and overlap matrices, which can be efficiently evaluated by a suitable Monte Carlo scheme [47, 48]. Finally, assuming that all states are properly normalized, the out-of-plane dynamical structure factor is given by:

$$S^z(\mathbf{q},\omega) = \sum_n |\langle \Psi_n^q | S_q^z | \Psi_0 \rangle|^2 \delta(\omega - E_n^q + E_0), \tag{13}$$

where $S_q^z = \frac{1}{\sqrt{N}} \sum_j e^{i\mathbf{q}\cdot\mathbf{r}_j} S_j^z$ ($\mathbf{r}_j$ being the coordinate of site $j$); $E_0$ is the variational ground-state energy of $|\Psi_0\rangle$ defined in Eq. (7), $E_n^q$ are the energies of the excited states $|\Psi_n^q\rangle$ for the momentum $\mathbf{q}$, as obtained from Eqs. (9) and (10).

Following a similar construction, we can also define spinon excitations with $S^z = 1$, and a different set of excited states and energies, which we still denote as $|\Psi_n^q\rangle$ and $E_n^q$, respectively, for simplicity of notation. By means of this variational set of excited states one can compute the in-plane dynamical structure factor:

$$S^\pm(\mathbf{q},\omega) = \sum_n |\langle \Psi_n^q | S_q^+ | \Psi_0 \rangle|^2 \delta(\omega - E_n^q + E_0). \tag{14}$$

The technical details for calculation of $S^\pm(\mathbf{q},\omega)$ are reported in Appendix C.

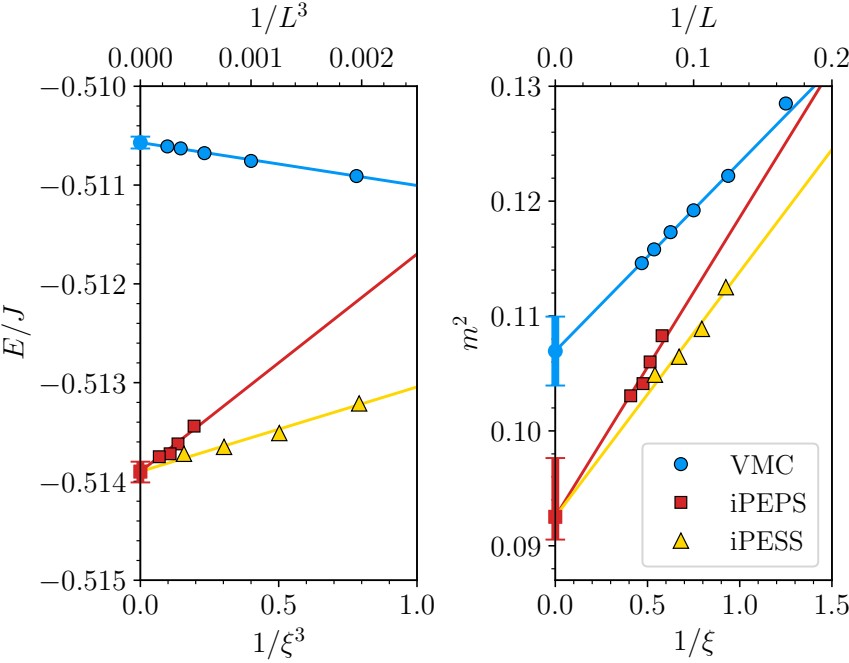

Figure 6: Thermodynamic extrapolations for VMC and TN (both iPEPS and iPESS) data for the ground-state energy per site (left panel) and the antiferromagnetic order parameter (squared) at $\mathbf{Q} = (0,0)$ (right panel), for $J_D/J = 0.25$. VMC results are obtained on $3 \times L \times L$ clusters and extrapolations are performed by using standard finite-size analysis for $L \to \infty$ (as shown in the upper $x$-axis). TN results are obtained by fixing the bond dimension $D$, evaluating the correlation length $\xi$, and then performing the finite-correlation-length scaling for $\xi \to \infty$ (as shown in lower $x$-axis).

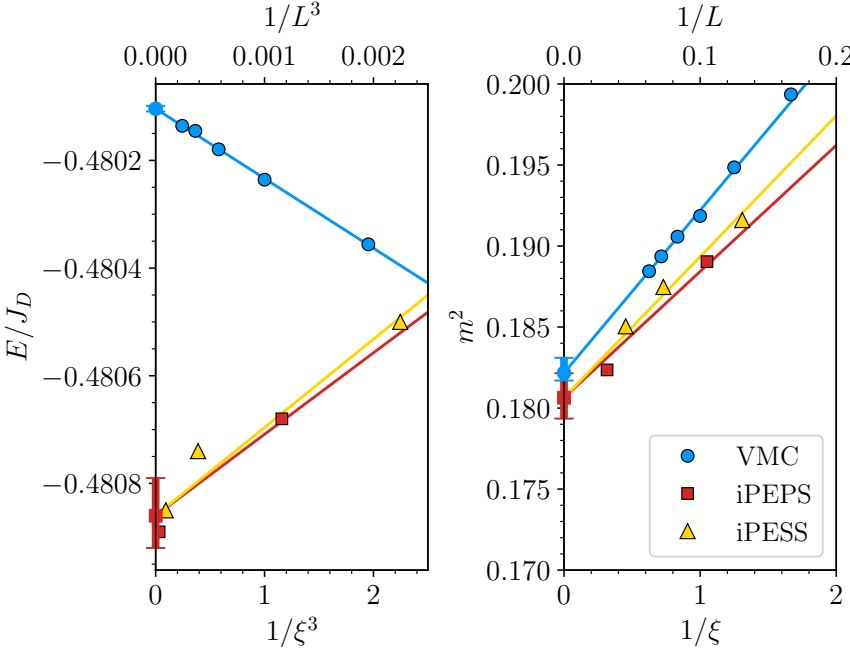

Figure 7: The same as in Fig. 6 for $J = 0$ and $J_D = 1$, i.e., the model with only DM interaction.

# 3 Results

In this section, we discuss the numerical results that have been obtained within TN and VMC approaches. First, we show the ground-state properties, focusing on the magnetization curve when $J_D/J$ is varied from 0 to 0.25 (suitable for $YCu_3(OH)_6Cl_3$), including $J_D/J = 0.18$ (suitable for $Cs_2Cu_3SnF_{12}$). Then, we present the results for the dynamical spin correlations, as obtained within the VMC technique. In particular, we concentrate on the case with $J_D/J = 0.18$, which allows us to highlight the remarkable similarities between the experimental outcome of Ref. [23] and the spectral functions of the Hamiltonian (1).

## 3.1 Static properties: energy and magnetization

The results of the squared magnetization $m^2$ at $\mathbf{Q} = (0, 0)$ for increasing values of $J_D/J$ are reported in Fig. 5, comparing VMC and TN results. Within VMC, we compute $m^2$ by evaluating the spin-spin correlations at the maximum distance for various cluster sizes $L$ and then perform the extrapolation $L \to \infty$. Within iPEPS and iPESS, instead, we measure the local magnetization $m^2 = |\langle \vec{S} \rangle|^2$ (which is identical on every site with high precision) for different values of the bond dimension $D$. Then we perform the finite-correlation-length scaling, imposing the same extrapolated value for iPEPS and iPESS, which helps reducing the error in the thermodynamic estimate. Technical details have been discussed in Section 2. The transition between the magnetically disordered regime and the magnetically ordered phase is found to be in excellent agreement between VMC and TN techniques. Indeed, the transition point is estimated at $J_D/J = 0.030(5)$ by VMC and $0.040(5)$ by iPEPS and iPESS. We mention that separate fits of iPESS and iPEPS datasets locate the transition at $J_D/J = 0.04$-$0.05$, in agreement with the estimate obtained by combining the data in a single extrapolation. Similarly to the results of Ref. [26], our estimate for the $J_D/J$ value at the transition is considerably smaller than the one reported in Ref. [24] ($J_D/J \approx 0.1$). The latter is obtained by exact diagonalization on small clusters (up to 36 sites) and the extrapolation of the order parameter to the thermodynamic limit is affected by strong finite size effects, especially close to the phase transition. According to the VMC wave function, the magnetically disordered phase found for $0 < J_D/J \lesssim 0.03$ is a spin liquid state with a finite gap in the spinon spectrum (see Appendix B). The spinon gap closes in the Heisenberg limit ($J_D = 0$) where the variational state reduces to the $U(1)$ Dirac spin liquid [60, 61].

Close to the transition point, the magnetizations obtained with VMC and TN are compatible within the errorbar, except for $J_D/J = 0.04$, for which the TN estimation of $m^2$ is very small, while a finite value is obtained within VMC. For larger values of the DM interaction, i.e., $J_D/J \gtrsim 0.15$, VMC and TN give comparable values of $m^2$, the maximum difference being about 20%. Finally, for very large values of $J_D/J$, the agreement between VMC and TN is again excellent. For example, for the pure DM model with $J = 0$ (and $J_D = 1$), the energy difference between the two methods is smaller than $10^{-3}J_D$ and the estimated values of the order parameter, $m^2 \approx 0.181$ (TN) and $m^2 \approx 0.182$ (VMC), are compatible within errobars. These values of $m^2$ are also in good agreement with the one reported in Ref. [24], once the latter is corrected by a factor $1/2$ due to a different definition of the order parameter. The actual comparison between TN and VMC is highlighted in Figs. 6 and 7, where we report both energy and magnetization (squared) for $J_D/J = 0.25$ and $J = 0$, respectively. We remark that, in all cases we examined, the largest difference between the extrapolated energies $\Delta E = E_{VMC} - E_{TN}$ is small and it is found in the Heisenberg model limit (i.e., $J_D = 0$), where $\Delta E/J \approx 0.007$.

## 3.2 Dynamical properties: spin correlation functions

Let us now focus on the low-energy spectrum as detected by the dynamical spin structure factor. Since, in presence of a finite DM interaction the SU(2) spin symmetry is explicitly broken, we consider both in-plane $S^{\pm}(\mathbf{q}, \omega)$ and out-of-plane $S^z(\mathbf{q}, \omega)$ responses. Before discussing the VMC results, it is instructive to look at the outcome of the linear spin-wave (LSW) approach, i.e., the simplest approximation for the low-energy spectrum of the ordered phase [64] (white lines in Fig. 8, for $J_D/J = 0.18$). Here, the spectral weight is concentrated on three magnon modes. One of them, representing the Goldstone mode, is gapless at $\Gamma$ (i.e., the center of the Brillouin zone) and $\Gamma'$ (i.e., the midpoint of the edges of the extended Brillouin zone). Its maximal intensity is observed in the in-plane response at $\Gamma'$, see Fig. 8 for $J_D/J = 0.18$. The other two magnon branches are gapped, one having a completely flat dispersion, and show the largest weight around the $\Gamma'$ point in the out-of-plane correlations. We mention that the LSW approximation can be reproduced within our VMC approach by taking no spinon hopping in the auxiliary Hamiltonian that defines the unprojected wave funtion (see Section 2), such that its ground state becomes a spin produt state in real space [65] (for that, the presence of the spin-spin Jastrow factor is fundamental to reproduce the correct magnon dispersions). In order to go beyond the LSW approach and assess both the renormalization of magnon branches and the presence of a continuum at low energies, we employ the dynamical VMC previously discussed.

The variational results for both in-plane and out-of-plane dynamical correlations are shown in Fig. 8, for $J_D/J = 0.18$, along the high-symmetry path $\Gamma - \Gamma' - K - \Gamma$ (qualitatively similar results are obtained for $J_D/J = 0.25$). Additional spectral functions, close to the the quantum phase transition, are reported in Appendix C. Within the low-energy portion of the VMC spectrum, the three magnon branches can be recognized thanks to their different intensities in the $S^z(\mathbf{q}, \omega)$ and $S^{\pm}(\mathbf{q}, \omega)$ channels, which can be matched to the LSW predictions. The first aspect that is worth noticing when comparing the outcome of the variational calculations with the LSW theory is the sensible downward renormalization of the magnon dispersion, which has been also discussed in the analysis of experimental measurements [66]. Indeed, both the gapless and the gapped dispersive branches are squeezed in energy, with a corresponding reduced velocity of the Goldstone mode around $\Gamma'$; by contrast, the flat magnon branch of the LSW theory acquires a (small) dispersion in the VMC spectrum, as can be seen in the in-plane response. Most importantly, the variational approach is able to describe a broad continuum above the magnon branches, i.e., a dense bunch of excitations with moderate weights which extends up to $\omega/J \approx 3.5$. Within the continuum, a relatively intense and weakly dispersing (damped) mode exists around $\Gamma'$, right above the three magnon excitations, clearly visible in the in-plane structure factor. This is a genuine hallmark of the dynamical structure factor when magnetic order develops beyond the quantum critical point; indeed, in the Heisenberg model with no DM interaction, the spectrum looks very broad and uniform, with no particular features in the continuum, see Fig. 9. The energy of the damped mode is $\omega \approx J$, very similar to the one observed within the inelastic neutron scattering in $Cs_2Cu_3SnF_{12}$ [23]. The outcome clearly suggests that the continuum is not featureless, but instead possesses non-trivial aspects that go beyond what can be captured by the simple LSW approximation.

The similarity between our VMC results and the experimental outcome is also evident from Fig. 10, where we report the total dynamical structure factor $S^{tot}(\mathbf{q}, \omega) = S^z(\mathbf{q}, \omega) + S^{\pm}(\mathbf{q}, \omega)$ along the cuts in momentum space considered in Ref. [23]. For a direct comparison, we take $J = 12.8$ meV, as estimated within the experimental work. Even though only a few $\mathbf{q}$ points are available along these cuts in the $3 \times 12 \times 12$ cluster, the global picture emerges. The most intense signal appears at very low energy around the $\Gamma'$ point [obtained for $K = 1/2$ and $H = 1/2$ in the $x$ axes of Fig. 10(a)], corresponding to the gapless Goldstone mode; the other two (gapped) magnons are mostly visible around $\Gamma'$, where their energies almost coincide,

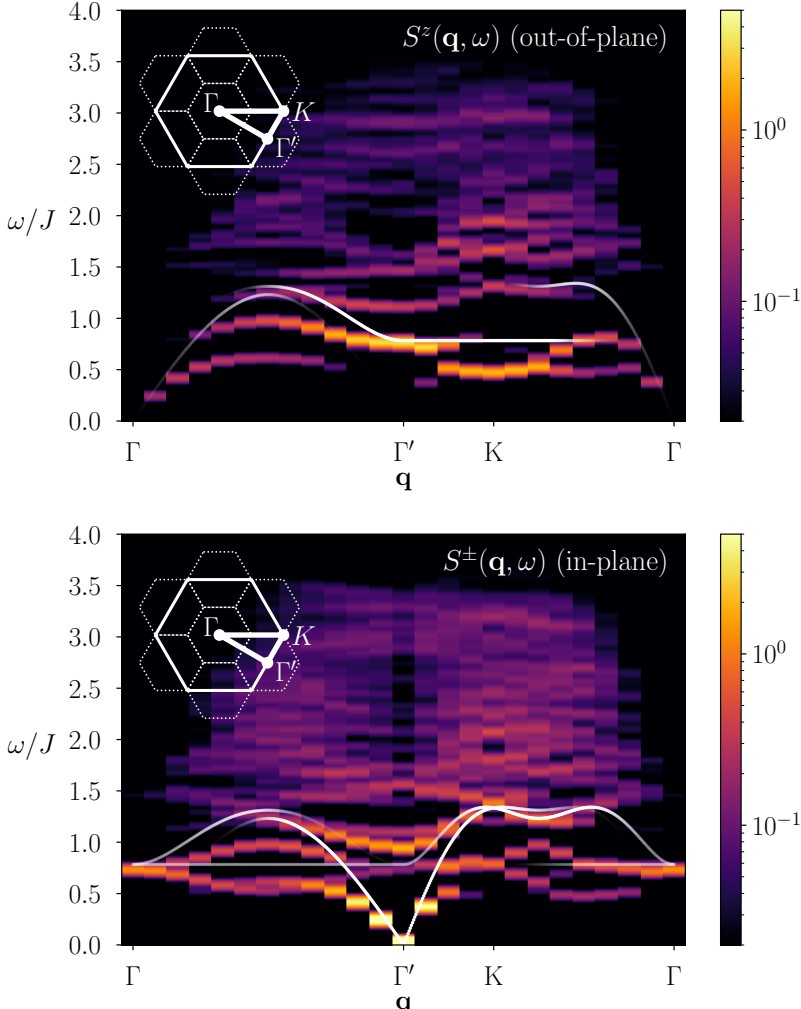

Figure 8: Out-of-plane [$S^z(\mathbf{q}, \omega)$, upper panel] and in-plane [$S^\pm(\mathbf{q}, \omega)$, lower panel] spin dynamical structure factor, as computed by dynamical VMC, see Eqs. (13) and (14), for $J_D/J = 0.18$. Calculations are performed on the $3 \times 12 \times 12$ cluster. LSW results for the three magnon modes are also shown for comparison (the intensity of the signal is proportional to the thickness of the white lines; notice that a different scale for the in-plane and out-of-plane components is used) [64]. The inset in the upper left corner of the figures shows the first and extended Brillouin zones of the kagome lattice (with dashed and solid lines, respectively), and the high symmetry points $\Gamma$ and $\Gamma'$.

i.e., $\omega \approx 10$ meV. On top of these magnon modes, an additional branch is also visible, with the maximum intensity at $\omega \approx 13$ meV (again at $\Gamma'$) and an upturning dispersion. This latter feature can be directly related to the one reported between 12 and 14 meV in inelastic neutron scattering experiments [23]. However, the relative intensities of the experimental peaks are not fully reproduced; in particular, at the $M$ point, the high-energy peak at $\omega \approx 15$ meV has a relatively large intensity, which is almost equal to the one of the magnons at $\omega \approx 7$ meV, see Fig. 10(b). By contrast, in experiments this feature is not observed. In general, a close comparison between the experimental results and numerical calculations is very difficult and goes beyond the scope of the present work. Indeed, many details (e.g., further super-exchange couplings, anisotropies, and disorder effects) may affect the outcome. In addition,

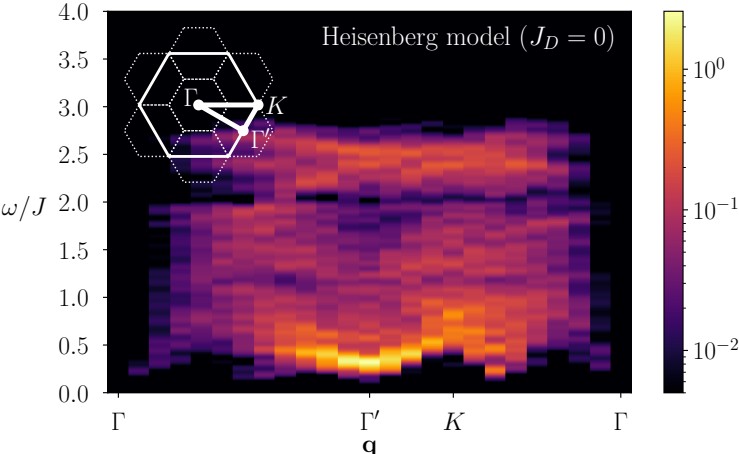

Figure 9: $S^z(\mathbf{q}, \omega)$ spin dynamical structure factor, as computed by the VMC approach, see Eqs. (13), for the Heisenberg model with $J_D = 0$. Calculations are performed on the $3 \times 12 \times 12$ cluster.

within our variational approach, the number of excited states is limited by the cluster size; as a consequence, providing a reliable estimate for the boundaries of the continuum is not possible. Nevertheless, our results strongly suggest that the broad peak observed at $\Gamma'$ is located above the magnon modes and can be thus expected to lie inside the continuum of excitations. The presence of an intense signal above the three magnon modes is a genuine feature of the model that cannot be described within a single-magnon picture.

## 4 Discussion

By employing a combination of VMC and TN methods, we have shown that the spin-1/2 kagome antiferromagnet develops a finite $\mathbf{Q} = (0, 0)$ magnetic order when an out-of-plane DM interaction $J_D$ exceeds a small fraction $0.03-0.04$ of the nearest-neighbor super-exchange $J$. In view of these results, a simple model with only $J$ and $J_D$ may not be suitable for the description of the *Herbertsmithite* material, since the value of $J_D$ interaction has been estimated to be above the critical value found here, while the compound does not show any sign of magnetic ordering down to the lowest temperature. In this regard, additional ingredients (e.g., substitutional disorder) should be included for a proper description of this material. By contrast, we believe that a model with only $J$ and DM interaction $J_D$ (above the critical value) may well capture the main features (at least at low temperature) of both $Cs_2Cu_3SnF_{12}$ and $YCu_3(OH)_6Cl_3$ materials. In particular, the dynamical structure factor of this minimal model can be directly compared to the results of inelastic neutron-scattering experiments on $Cs_2Cu_3SnF_{12}$. Within the magnetically ordered phase, besides the magnon branches, the existence of additional damped mode in the continuum is reported, in close similarity to what has been recently detected in $Cs_2Cu_3SnF_{12}$ [23]. Open questions remain to fully characterize the magnetically disordered regime. Within the Gutzwiller-projected fermionic approach, a triplet hopping develops as soon as $J_D$ is finite, opening a gap in the spinon spectrum. Nevertheless, the incipient magnetic phase does not allow the stabilization of a spin liquid state with a sufficiently large gap to be detected within the available numerical calculations, whose momentum resolution is limited by finite size effects. Understanding whether a detectable gapped state may exist when suitable perturbations are added on top of the Heisenberg model is an important issue that should be further addressed in future investigations.

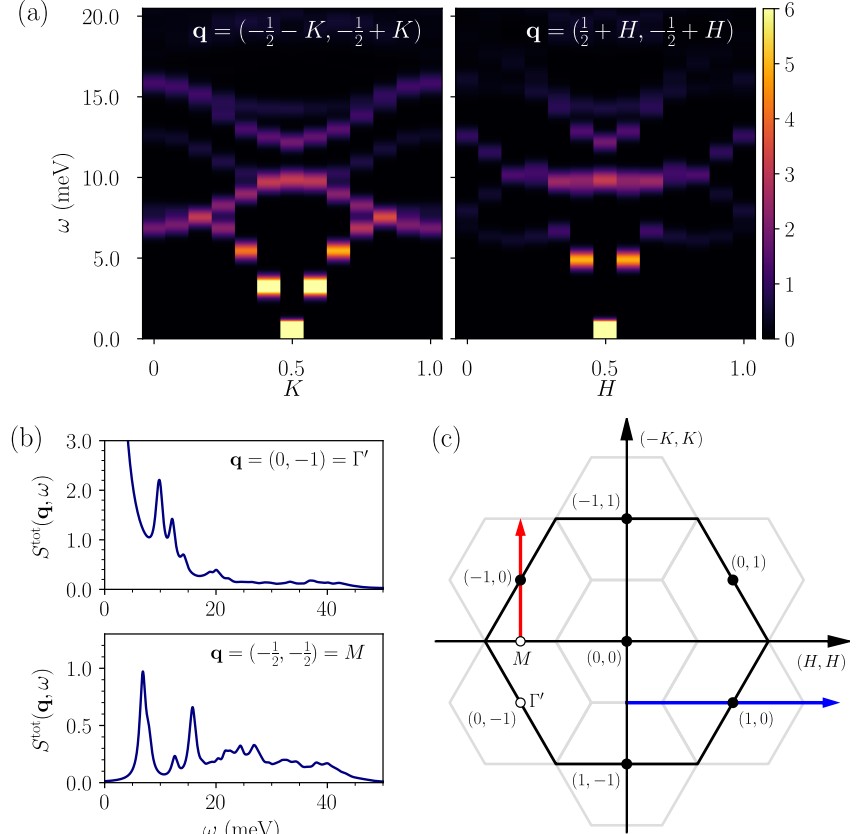

Figure 10: Panel (a): Total spin dynamical structure factor $S^{\text{tot}}(\mathbf{q}, \omega) = S^z(\mathbf{q}, \omega) + S^{\pm}(\mathbf{q}, \omega)$ for $J_D/J = 0.18$ on the $3 \times 12 \times 12$ cluster, along the cuts in momentum space shown in panel (c), reproducing the ones used in Ref. [23]. Here, the super-exchange coupling has been fixed to $J = 12.8$ meV and $J_D/J = 0.18$, as estimated in Ref. [23] for $Cs_2Cu_3SnF_{12}$. Panel (b): The dynamical structure factor as a function of $\omega$ at $\Gamma'$ and $M$ points (a broadening of $\sigma = 0.05J$ has been introduced to have a smooth curve). Panel (c): first (grey hexagons) and extended (black hexagon) Brillouin zones of the kagome lattice. The red and blue arrows depict the paths along $\mathbf{q} = (-\frac{1}{2} - K, -\frac{1}{2} + K)$ and $\mathbf{q} = (\frac{1}{2} + H, -\frac{1}{2} + H)$ considered in panel (a). The empty dots indicate the $\Gamma'$ and $M$ points considered in panel (b).

# Acknowledgements

We thank F. Bert, K. Riedl and C. Wang for useful discussions. Y.I. acknowledges support from DST through the grants SRG No. SRG/2019/000056, MATRICS No. MTR/2019/001042, and CEFIPRA No. 64T3-1, ICTP through the Associates Programme and the Simons Foundation through grant number 284558FY19. This research was supported in part by the National Science Foundation under Grant No. NSF PHY-1748958, IIT Madras through the QuCenDiEM group (Project No. SB20210813PHMHRD002720), FORG group (Project No. SB20210822PHMHRD008268), the International Centre for Theoretical Sciences (ICTS), Bengaluru, India during a visit for participating in the program "Frustrated Metals and Insulators" (Code: ICTS/frumi2022/9), the European Research Council (ERC) under the European Union's Horizon 2020 research and innovation programme (grant agreement No 101001604), TNTOP ANR-18-CE30-0026-01 grant awarded from the French Research Council. This work

was also granted access to the HPC resources of CALMIP supercomputing center under the allocations 2017-P1231 and 2021-P0677. Y.I. acknowledges the use of the computing resources at HPCE, IIT Madras. F.F. acknowledges financial support from the Alexander von Humboldt Foundation through a postdoctoral Humboldt fellowship and by the Deutsche Forschungsgemeinschaft (DFG, German Research Foundation) for funding through TRR 288 – 422213477 (project A05). F.F. is grateful to LPT Toulouse for the invitation which led to the beginning of this project. F.F. thanks IIT Madras for funding a one-month stay through an International Visiting Postdoctoral Travel award which facilitated completion of this research work.

# A    Details on the tensor network calculations

## A.1    Extrapolations as a function of the CTM environment bond dimension $\chi$

For each fixed bond dimension $D$, the variational optimization are performed with different CTM environment bond dimensions $\chi$. Then, observables are evaluated for each value $\chi$ and finally etrapolated for $\chi \to \infty$. We show the case with $J_D/J = 0.02$ in Fig. 11. In practice, local observables (e.g., energy and magnetization) converge very rapidly, with small linear corrections in $1/\chi$; instead, the correlation length displays slightly larger corrections, which typically require a quadratic fitting.

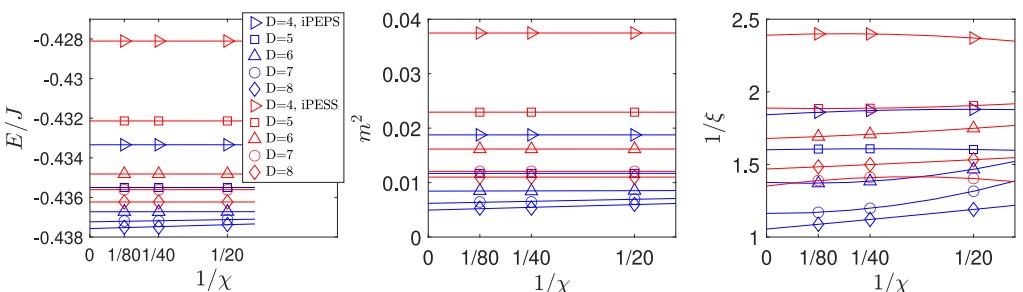

Figure 11:   Extrapolations of the energy, magnetization (square), and correlation length as a function of the CTM environment dimension $\chi$ for $J_D/J = 0.02$.

## A.2    Analysis of *Ansätze* and optimizations

Here, we analyze the TN data in terms of *Ansätze* and optimizations. First, we consider the variational optimization, as adopted in this work, and compare energies and magnetizations for iPESS and iPEPS *Ansätze*, see Fig. 12. For every fixed value of $D$, iPEPS has lower energy and smaller magnetization than iPESS. This is due to the fact that the latter one is more constrained. On the other hand, symmetry properties of iPESS are better preserved, since iPESS treats down-pointing and up-pointing triangles equally, while in iPEPS only the entanglement inside the down-pointing triangle is restricted by finite $D$ (similar to the 9-iPESS *Ansatz* defined in Ref. [40]). In the worst case (i.e., within the spin liquid regime), the difference of the energy contributions coming from different bonds is about 1% for the iPEPS with $D = 7$ (the point-group symmetries are expected to be restored only for $D \to \infty$).

We now briefly discuss the comparison between variational and simple update (SU) method to optimize the TN states. Within the former approach, the tensors of iPESS and iPEPS states are optimized by using gradients of the total energy, taking into account the global environment. By contrast, the SU approach ignores the non-local environment tensors during optimization and is only applicable to the iPESS *Ansatz* [26]. The comparison of these methods for energies and magnetizations is shown in Fig. 12. For each bond dimension $D$, the

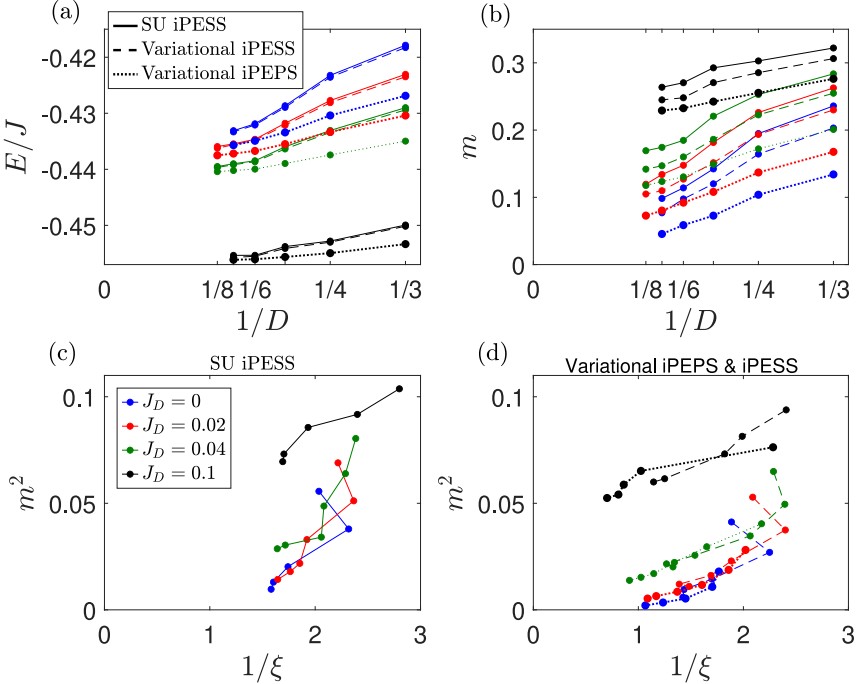

Figure 12: Comparison between SU iPESS (solid line), variational iPESS (dashed line) and variational iPEPS (dotted line) for different values of $J_D/J$ and $\chi = 80$. Panels (a) and (b): energies and magnetizations as a function of $1/D$. Panels (c) and (d): square of magnetizations as a function of $1/\xi$ for SU and variational optimizations.

two optimization techniques give similar energies (the variational one being slightly lower), but within the SU approach the magnetization is overestimated by roughly 10% and the correlation length is considerably underestimated. The good quality provided by the variational optimization is particularly important when performing scalings. Indeed, while the extrapolations of $m^2$ are quite smooth within the variational approach, the behavior obtained within the SU technique is more erratic and may lead to unphysical negative values for the extrapolated quantities.

# B  Spin liquid wave function of VMC

The Gutzwiller-projected wave functions employed in the VMC approach are defined by means of the auxiliary Hamiltonian (6), which contains hopping terms and a Zeeman field that induces $\mathbf{Q} = (0,0)$ magnetic order in the $XY$ plane. For $J_D/J \lesssim 0.03$, the Zeeman field parameter $h$ is found to vanish in the thermodynamic limit, signalling the onset of a spin-liquid phase, which is described by the Hamiltonian with only hopping terms. The optimal variational *Ansatz* contains a real singlet hopping ($\chi_s$) and an imaginary triplet hopping ($\chi_t$) at first-neighbors. The sign structure of the hoppings is dictated by the projective symmetry group [33] and is schematically represented in Fig. 13. In the Heisenberg limit, $J_D = 0$, $\chi_t$ vanishes (exactly) upon minimization of the variational energy and the auxiliary Hamiltonian reduces to the one of the $U(1)$ Dirac spin-liquid state [27, 60, 61], with gapless Dirac points in the spinon spectrum. On the other hand, for any $J_D/J > 0$, the variational optimization of the wave function yields a finite value of the triplet hopping $\chi_t$ (see Fig. 13), which opens a gap in the unprojected

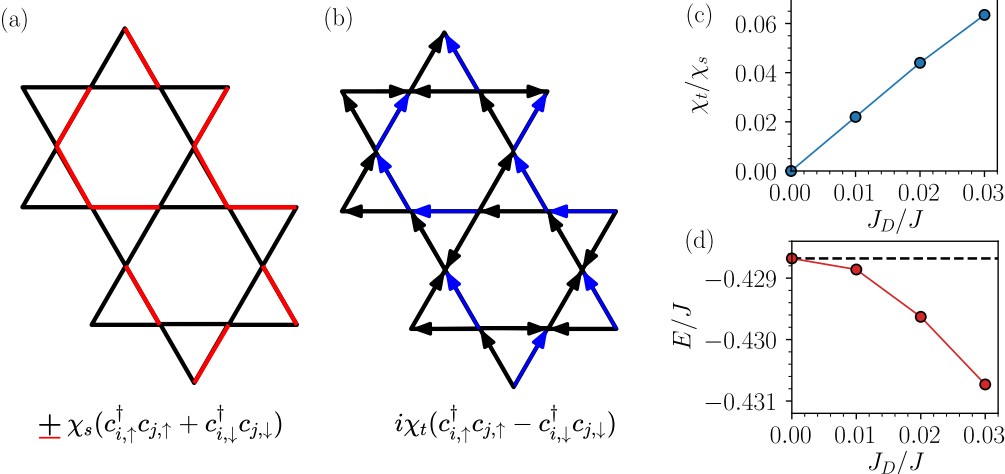

(a) $\pm \chi_s(c^{\dagger}_{i,\uparrow}c_{j,\uparrow} + c^{\dagger}_{i,\downarrow}c_{j,\downarrow})$

(b) $i\chi_t(c^{\dagger}_{i,\uparrow}c_{j,\uparrow} - c^{\dagger}_{i,\downarrow}c_{j,\downarrow})$

Figure 13: Spin liquid wave function used within the VMC approach. Panel (a): sign pattern of the real-valued singlet hoppings of the auxiliary Hamiltonian $\mathcal{H}_0$, with black (red) bonds denoting $\chi_s$ ($-\chi_s$) terms. This variational *Ansatz* corresponds to the Dirac spin liquid state [27, 60, 61]. Panel (b): sign pattern of the imaginary triplet hoppings of the auxiliary Hamiltonian $\mathcal{H}_0$. The arrows from $i \to j$ represent the triplet term $i\chi_t(c^{\dagger}_{i,\uparrow}c_{j,\uparrow} - c^{\dagger}_{i,\downarrow}c_{j,\downarrow})$, which is odd under the exchange of $i$ and $j$ sites. Note that the orientations of the black (blue) arrows is equal (opposite) to the ones of the DM interaction (see Fig. 1). Panel (c): optimal values of the ratio $\chi_t/\chi_s$ for the spin-liquid *Ansatz* in the thermodynamic limit. Panel (d): thermodynamic limit of the variational energy of the spin liquid state (with no Zeeman field and no Jastrow factor). The dashed line indicates the energy of the Dirac spin liquid state, which is independent on the value of $J_D$.

spinon spectrum and provides a variational energy gain with respect to the Dirac spin-liquid state (whose energy is independent on the value of $J_D$). Although the fermionic Hamiltonian is gapped in the thermodynamic limit, a precise estimation of the excitation gap in the spectrum of the spin Hamiltonian can be obtained only by a numerical calculation with the Gutzwiller-projected states, which is computationally demanding and left for future work. We also note that, unfortunately, we cannot give definitive statemets on the nature of the spin-liquid regime (e.g., whether it is a $U(1)$ or $\mathbb{Z}_2$ state), because additional superconducting pairings can be stabilized in the auxiliary Hamiltonian upon numerical optimizations [33], although they do not provide an appreciable energy gain.

## C   Dynamical variational Monte Carlo

As discussed in the main text, we compute both in-plane $[S^{\pm}(\mathbf{q}, \omega)]$ and out-of-plane $[S^z(\mathbf{q}, \omega)]$ dynamical structure factor by VMC. A detailed discussion of the method to compute $S^z(\mathbf{q}, \omega)$ is given in Refs. [46–48]. Here, we present an extension of the dynamical VMC technique for the calculation of the $S^{\pm}(\mathbf{q}, \omega)$ component.

Starting from the optimal fermionic wave function $|\Phi_0\rangle$, we define a set of projected particle-hole excitations with momentum $\mathbf{q}$ as follows

$$|q;R,a;b\rangle = \mathcal{J}\mathcal{P}_G\mathcal{P}_{S^z=1}\sum_{R'}e^{i\mathbf{q}\cdot\mathbf{R'}}c^{\dagger}_{R+R',a,\uparrow}c_{R',b,\downarrow}|\Phi_0\rangle. \tag{C.1}$$

Here and in the following, we label lattice sites by specifying their Bravais lattice vector $\mathbf{R}$ (or $\mathbf{R}'$) and their sublattice index, denoted by Latin letters, e.g., $a$ (or $b$). Then, we construct variational excited states for the system by taking linear combinations of the particle-hole excitations $|q; R, a; b\rangle$, namely

$$|\Psi_n^q\rangle = \sum_R \sum_{a,b} B_{R,a;b}^{n,q} |q; R, a; b\rangle, \tag{C.2}$$

where $n$ is an integer label. The coefficients $B_{R,a;b}^{n,q}$ are obtained by the Rayleigh-Ritz method, i.e., by solving the generalized eigenvalue problem

$$\sum_{R',a',b'} H_{R,a;b|R',a';b'}^q B_{R',a';b'}^{n,q} = E_n^q \sum_{R',a',b'} O_{R,a;b|R',a';b'}^q B_{R',a';b'}^{n,q}, \tag{C.3}$$

for each desired momentum $\mathbf{q}$. Here, $E_n^q$ are the energies of the excitation $|\Psi_n^q\rangle$, and

$$H_{R,a;b|R',a';b'}^q = \langle q; R, a; b|\mathcal{H}|q; R', a'; b'\rangle, \tag{C.4}$$

$$O_{R,a;b|R',a';b'}^q = \langle q; R, a; b|q; R', a'; b'\rangle, \tag{C.5}$$

are the Hamiltonian and overlap matrices, respectively. Their entries are computed stochastically by a suitable Monte Carlo scheme, which is outlined in the following for the case of the overlap matrix (the Hamiltonian matrix can be treated analogously). By inserting a resolution of the identity over fermionic configurations $\{|x\rangle\}$, we can write

$$O_{R,a;b|R',a';b'}^q = \sum_x \langle q; R, a; b|x\rangle\langle x|q; R', a'; b'\rangle. \tag{C.6}$$

Due to the presence of $\mathcal{P}_G$ and $\mathcal{P}_{S^z=1}$ in the definition of $|q; R, a; b\rangle$ states [see Eq. (C.1)], the states $\{|x\rangle\}$ are constrained to one-fermion-per-site configurations with $S^z = 1$. Therefore, we can compute the entries of the overlap matrix by a Metropolis algorithm in which we sample the Hilbert space according to the probability function $|\langle x|\Psi_0'\rangle|^2/\|\Psi_0'\|^2$, where $|\Psi_0'\rangle = \mathcal{J}\mathcal{P}_{S_z=1}\mathcal{P}_G|\Phi_0\rangle$ and $\|\Psi_0'\|^2 = \langle\Psi_0'|\Psi_0'\rangle$. Thus, we can evaluate the rescaled overlap matrix

$$\tilde{O}_{R,a;b|R',a';b'}^q = \frac{O_{R,a;b|R',a';b'}^q}{\|\Psi_0'\|^2} = \sum_x \frac{|\langle x|\Psi_0'\rangle|^2}{\|\Psi_0'\|^2}\left[\frac{\langle q; R, a; b|x\rangle}{\langle\Psi_0'|x\rangle}\frac{\langle x|q; R', a'; b'\rangle}{\langle x|\Psi_0'\rangle}\right]. \tag{C.7}$$

An analogous formula applies for the calculation of a rescaled Hamiltonian matrix

$$\tilde{H}_{R,a;b|R',a';b'}^q = \frac{H_{R,a;b|R',a';b'}^q}{\|\Psi_0'\|^2} = \sum_x \frac{|\langle x|\Psi_0'\rangle|^2}{\|\Psi_0'\|^2}\left[\frac{\langle q; R, a; b|x\rangle}{\langle\Psi_0'|x\rangle}\frac{\langle x|\mathcal{H}|q; R', a'; b'\rangle}{\langle x|\Psi_0'\rangle}\right]. \tag{C.8}$$

Then, we solve the generalized eigenvalue problem of Eq. (C.3) with the rescaled matrices. The resulting $B_{R,a;b}^{n,q}$ coefficients are normalized such that

$$\sum_{R,R'}\sum_{a,a'}\sum_{b,b'}[B_{R,a;b}^{n,q}]^*\tilde{O}_{R,a;b|R',a';b'}^q B_{R',a';b'}^{m,q} = \delta_{n,m}. \tag{C.9}$$

As a consequence, the variational excited states satisfy $\langle\Psi_n^q|\Psi_m^q\rangle = \|\Psi_0'\|^2\delta_{n,m}$.

The excited state energies $\{E_n^q\}$ enter the Lehmann representation of $S^\pm(\mathbf{q}, \omega)$, together with the ground state energy $E_0$ (computed in the $S_z = 1$ sector) and the spectral weights. The latter are evaluated as

$$\frac{\langle\Psi_n^q|S_q^+|\Psi_0\rangle}{\|\Psi_n^q\|\|\Psi_0\|} = \sum_x \frac{|\langle x|\Psi_0'\rangle|^2}{\|\Psi_0'\|^2}\left[\frac{\langle\Psi_n^q|x\rangle}{\langle\Psi_0'|x\rangle}\frac{\langle x|S_q^+|\Psi_0\rangle}{\langle x|\Psi_0'\rangle}\right]\frac{\|\Psi_0'\|}{\|\Psi_0\|}. \tag{C.10}$$

In practice, directly computing the factor $\|\Psi_0'\|/\|\Psi_0\|$ is an unfeasible task. Therefore, in the actual calculations, we sample only the quantity within square brackets in Eq. (C.10) and we correct the spectral weights *a posteriori* by enforcing the sum rule

$$\int d\omega \, S^{\pm}(\mathbf{q},\omega) = \frac{\langle\Psi_0|S_q^- S_q^+|\Psi_0\rangle}{\|\Psi_0\|^2}, \tag{C.11}$$

where the r.h.s. can be computed within standard VMC.

Finally, we note that, for momenta equivalent to $\mathbf{q} = (0,0)$ (e.g., $\Gamma$ and $\Gamma'$ points), we include an additional projector in the definition of the $|q;R,a;b\rangle$ states, which makes them orthogonal to the $S_z = 1$ variational ground state.

In conclusion, we observe that the main difference between the present Monte Carlo scheme and the one of Ref. [46] lies in the fact that here the $S_z = 1$ sector is sampled. Both

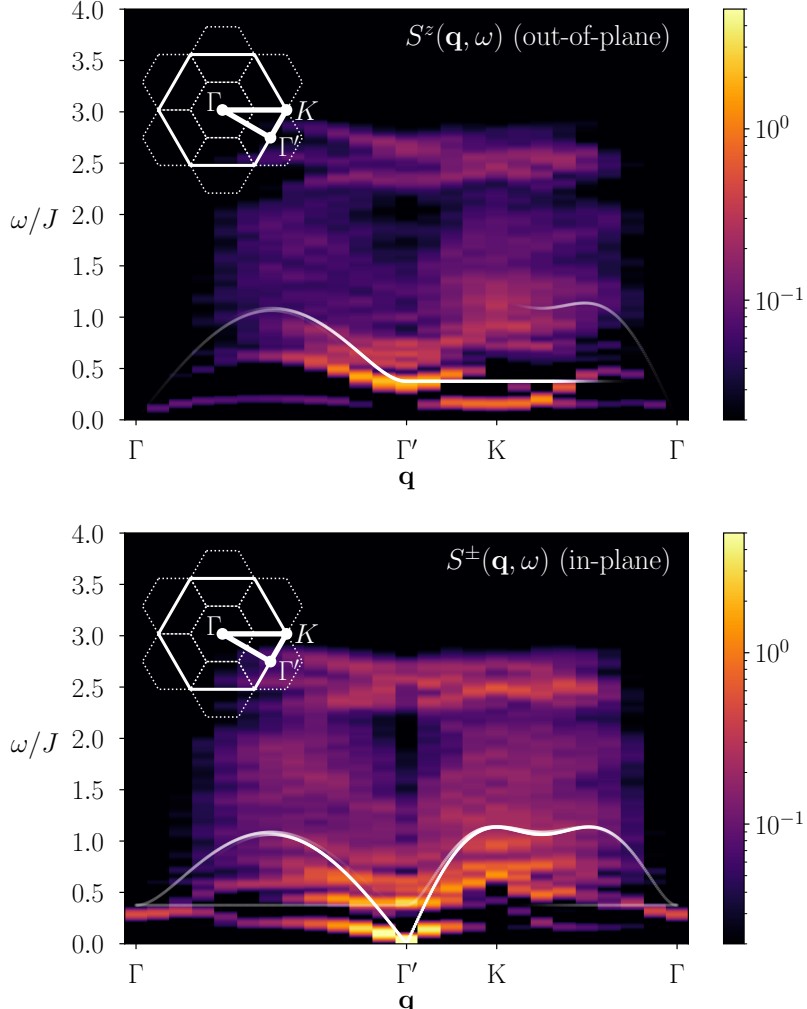

Figure 14: Out-of-plane [$S^z(\mathbf{q},\omega)$, upper panel] and in-plane [$S^{\pm}(\mathbf{q},\omega)$, lower panel] spin dynamical structure factor, for $J_D/J = 0.05$. Calculations are performed on the $3 \times 12 \times 12$ cluster. LSW results for the three magnon modes are also shown for comparison (the intensity of the signal is proportional to the thickness of the white lines) [64]. The inset in the upper left corner of the figures shows the first and extended Brillouin zones of the kagome lattice (with dashed and solid lines, respectively), and the high symmetry points $\Gamma$ and $\Gamma'$.

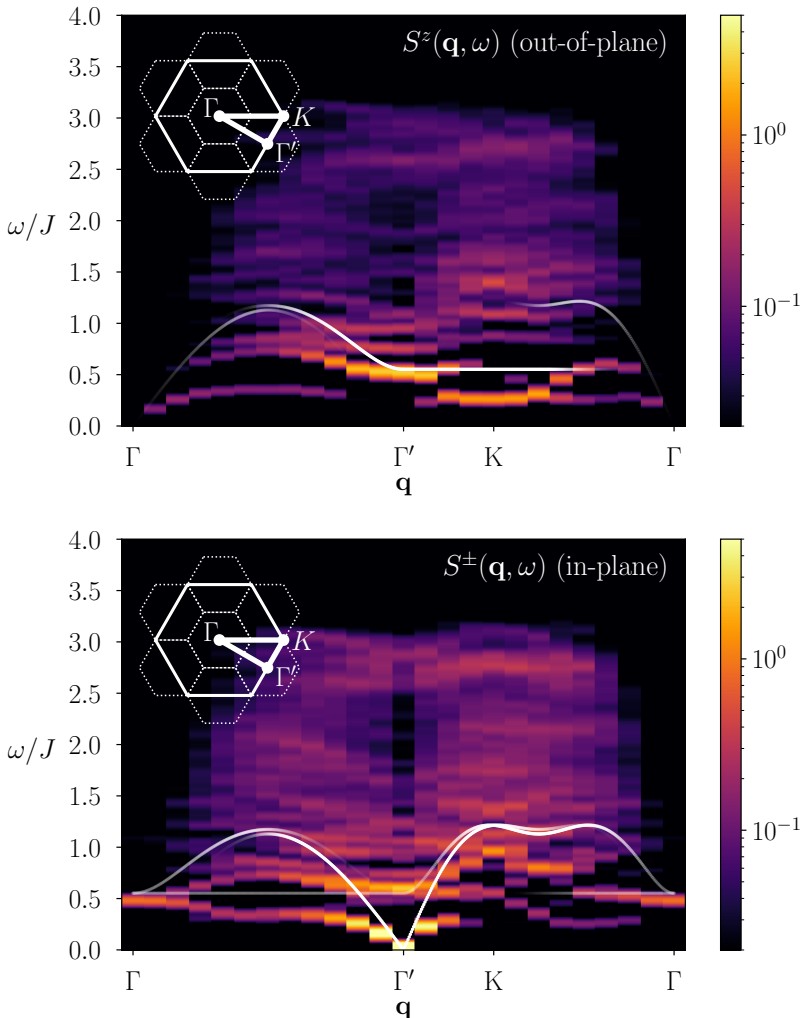

Figure 15: The same as in Fig. 14 for $J_D/J = 0.1$.

methods share a high degree of computational efficiency, since a single Monte Carlo run is sufficient to compute all the entries of the Hamiltonian and overlap matrices, and the spectral weights.

The results for $J_D/J = 0.05$ and 0.1 are reported in Figs. 14 and 15. By approaching the quantum phase transition to the spin-liquid regime, the whole lowest-energy magnon excitation is strongly renormalized towards smaller energies; in addition, the visible (damped) modes within the continuum lose spectral weight and the continuum becomes progressively broader upon decreasing the DM interaction. We note that our spectrum at $\Gamma'$ for $J_D/J = 0.05$ shares some similarities with the one of Ref. [36] at $J_D/J = 0.06$. However, the latter, computed by density-matrix renormalization group calculations on finite-width cylinders, strongly depends on the choice of boundary conditions along the rungs [50]. In the case of anti-periodic boundary conditions, the in-plane dynamical structure factor at $\Gamma'$ is dominated by an intense peak at $\omega \approx 0$, while the out-of-plane component is weaker and shows a peak around $\omega/J \approx 0.4$ [36]. This is comparable to our results in Fig. 14, although we ascribe the existence of the strong peak at $\omega \approx 0$ to the presence of magnetic order, contrary to the conclusions of Ref. [36].

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
