# Peer review of "Static and dynamical signatures of Dzyaloshinskii-Moriya interactions in the Heisenberg model on the kagome lattice"

_SciPost Physics, doi:SciPost Phys. 14, 139 (2023)_

## Round 1 · Referee Report · Philippe Mendels (Referee 1) · 2023-1-16

Strengths

1- Substantial improvement of the TNS approach on the role of Dzyaloshinkii-Moriya interaction in stabilizing the ground state of the kagome Heisenberg antiferromagnet 2- Comparison to existing experimental data

Weaknesses

Refine the comparison to the experimental data

Report

From my point of view, the paper provides a very interesting study of the impact of the DM interaction on the physics of KHAF, more specifically an upgrade of the method using TNS to predict the critical value beyond which a Néel order is stabilized. Early on predicted to set in above the D/J~0.08 critical value, the problem was reconsidered using tensor network states approach predicting an amazingly small value of 0.01, putting the emblematic spin liquid herbertsmithite well above that limit, then leading to the prediction of an ordered phase. The results provided by the authors basically using the same technique but with different optimization and extrapolation schemes yield a critical value around 0.04-0.05 J which lies in between the early prediction and that provided by the first TNS treatment of the problem. This a very precious result, all kagome QSL candidates have no bond-inversion symmetry and I do recommend the publication of this manuscript once other theorist referee(s) validates the authors’ numerical approach.
The authors compare their results to two ordered compounds and to herbertsmithite, one which has a quasi-perfect kagome geometry, the so-called Y-kapellasite and the other one which clearly shows a distorted kagome lattice, interaction-wise for which refined neutron scattering data have been recently published.
I have five small comments.
1. The D value for herbertsmithite (HBS) extracted from ESR data on powders has been scaled down after the first 0.08 J estimate by Zorko et al. (Ref.[17]) to be in between 0.04 and 0.08 J with a most likely value of 0.05 J [El Shawish et al., Phys Rev B. 81, 224421 (2010)], not so far but still slightly above the critical value found by the authors. Note that the value could be refined now that single crystals are available. In any case, the authors would predict that the QSL state should be gapped, whereas this does not seem to be the case [P. Khuntia et al., Nature Physics 16, 469-474 (2020)], an issue still debated; maybe I missed the point but it would be useful to have that value of the gap (possibly vs J_D/J), other compounds might show up. As a note other recent experimental data on HBS show that the QSL state is gapless but not a U(1) Dirac QSL [Phys. Rev. X 12, 011014] and this is in line with the statement of the authors that substitutional disorder on the inter-layer site impacts the nature of the QSL ground state.
2. Regarding q=0 ordered compounds, the authors also estimate the value of the moment in the ordered phase versus the J_D/J ratio. It seems to me that their values are close to the values found in the paper by Cepas et al. while the critical value of J_D/J differs substantially. Can they comment on this, and also give their value for J_D-> infty? The latter does not seem to differ from Cepas et al. Any good reason for this?
3. In line with the previous comment, the value of the moment found in the ordered phase versus the J_D/J ratio is worth a comparison to the experimental data. I note, that the value found for m^2 is consistent with that found for Y-kapellasite, following the authors of ref.[19] (see page 4). On the contrary, after a bibliography search, the m value for the Cs2Cu3SnF12 is announced to be 0.68(3) mu_B which seems to be well beyond the upper limit expected [K. Matan et al. Phys Rev B 99, 224404 (2019)]. In my opinion this underlines the shift from the n.n. ideal Heisenberg case which is documented in ref.[20]. This calls for caution when comparing the neutron dynamical structure factor to the experimental one. Looking at page 5 of ref.[20], it seems that the gap between the two upper branches is closer to zero than the one calculated by the authors. Any reason for that?
4. I feel that one interesting track, introduced in the context of the experimental studies on herbertsmithite [M. Jeong et al. , Phys. Rev. Lett. 107, 237201 (2011)] would be to include an additional Zeeman term close but below the D_J/J critical value. Can we expect a field induced ordering or not and what would be the critical field (given in J unit)? That might also help to settle the debate about HBS whether has some facets of the perfect KHAF or whether inter-layer magnetic disorder has a leading role in stabilizing the QSL phase.
5. Following the authors’ will to cite an NMR evidence for a gapless ground state in HBS (ref[15]) and respecting others’ point of view, I would suggest to add two references:
P. Khuntia et al., Nature Physics 16, 469-474 (2020)
J. Wang et al., Nat. Phys. 17, 1109–1113 (2021)

Requested changes

See report

1- Take into account the revised values of the critical D_J from the work by El Shawish et al. 2- Compare in more quantitative details to the phase diagram by Cepas et al. 3- Compare the predictions for the order parameter when the q=0 phase to the experimenatla data and comment about it 4- Maybe include the additional references from my report

  • validity: -
  • significance: top
  • originality: high
  • clarity: -
  • formatting: perfect
  • grammar: perfect

Author:  Francesco Ferrari  on 2023-02-20  [id 3379]

(in reply to Report 1 by Philippe Mendels on 2023-01-16)

We thank the referee for their assessment of our work and we reply to their questions, addressing the requested changes.

1-Following the suggestion of the referee, we have included the revised estimates of J_D/J for Herberthsmithite in the paper.

2-Concerning the location of the phase transition, we note that Lanczos calculations are limited to small clusters (up to 36 sites) and the extrapolation of the order parameter to the thermodynamic limit is affected by strong finite size effects, especially close to the phase transition. A considerably smaller value of J_D/J for the transition is corroborated also by recent tensor networks results of [C.-Y. Lee, B. Normand, Y.-J. Kao, Phys. Rev. B 98, 224414, (2018)]. We note that the extension of the magnetic region was also underestimated by Lanczos calculations in the case of the J1-J2 Heisenberg model on the square lattice [H. J. Schulz et al, J. Physique I 6, 675 (1996)], in comparison to more recent estimates [e.g. J. Hasik, D. Poilblanc, F. Becca, SciPost Phys. 10, 012 (2021)].

As far as the value of m^2 is concerned, we note that the estimator of Cepas et al yields a value that is twice the one obtained by us. We emphasize that the standard estimator for m^2 is |<S>|^2, namely the one used in our paper. Our calculations at J_D=1,J=0 give approximately m^2=0.181, which is comparable to the value reported by Cepas, once it is corrected by the aforementioned factor 1/2 (i.e., m^2=0.326/2=0.163). We added a brief discussion on these aspects in the paper.

3-In general, a quantitative comparison obtained by using a simplified model and actual materials is not an easy task. In this respect, the actual value of the magnetic moment may be affected by many details that are not included in the Heisenberg model (e.g., anisotropies in the spin exchange, presence of further couplings, and even multi-orbital effects). An indication of this difficulty is shown by the fact that the magnetic order for YCu3(OH)6Cl3 is smaller than the one for Cs2Cu3SnF12, although the latter is expected to have a smaller value of J_D. Similar caution must be taken into account when comparing spectral functions to neutron scattering results, however our
study shows that a *qualitative* comparison of the main features is possible and hints to the presence of a sizable Dzialoshinskii-Moriya interaction in Cs2Cu3SnF12. This is discussed in the new version of the manuscript.

4-Following the suggestion of the referee, we have included the references.

---

## Round 1 · Referee Report · Anonymous (Referee 3) · 2023-1-30

Strengths

-detailed computational study of a fundamental model of frustrated magnetism on the kagome lattice to account for the consequences of DM interactions
-complemental methods approach (VQMC and different TN approaches)
-a consistent phase diagram is obtained
-quantitative results for excitations as obtained for the dynamical structure factor
-a short comparison to relevant experimental results is provided

Weaknesses

-too little discussion about the nature of the observed "broad peak"
-too little discussion regarding earlier results on the critical DM interaction strength and the spectral properties

Report

The authors explore the effects of DM interactions on the static and dynamical properties of the kagome lattice spin-1/2 Heisenberg model, motivated by its relevance to several recently examined compounds. For their study they use a combination of VQMC and TN methods, which provides a consistent phase diagram exhibiting the transition from a quantum spin liquid to an ordered state for relatively weak DM interaction strength. Similar analysis has been performed previously, and the authors claim to have reached a quantitatively consistent phase diagram from their different methods and in addition also examine (by VQMC) the dynamical properties in terms of the dynamical spin structure factors. There they observe the relevant low-energy excitations and in addition to the multi-particle continuum identify a further characteristic "broad peak" feature beyond the low-energy regime.

The results of this analysis are relevant for the exploration of kagome-based frustrated magnetism, in particular in view of various recently explored compounds, and this aspect is well taken by the authors. The authors also provide a careful analysis of the various computational approaches they used, and the argue for a consistent picture to emerge. In general, such a detailed analysis of this basic quantum spin model featuring a quantum spin liquid phase is certainly of great interest also from a theoretical and methods perspective. The paper is also well structured and written, with clear figures provided.

I recommend to publish this manuscript after the requested changes, detailed below, have been implemented.

Requested changes

-VQMC results for J_D=0 are not shown in Fig. 4 and 5. This should be included or it should be explained, why it is missing.
-The authors should state how well the two TN approaches agree/disagree without fixing the extrapolated values of the order parameter between iPEPS and IPESS.
-The authors could estimate the boundaries of the continuum from their observed low-energy dispersion. This could be useful also for a more detailed discussion of the observed "broad peak". Is it possibly a bound state split off the continuum?
-It would be useful to discuss in more detail the discrepancies of the current results for the critical DM interaction strength with respect to easier reported values. Also, the authors should comment on the results from earlier studies of the spectral properties in Sec. 3.2 (thus far, earlier work is mentioned only in Sec. 1).

  • validity: top
  • significance: high
  • originality: good
  • clarity: high
  • formatting: excellent
  • grammar: excellent

Author:  Francesco Ferrari  on 2023-02-20  [id 3381]

(in reply to Report 3 on 2023-01-30)

We thank the referee for their assessment of our work and we reply to their questions, addressing the requested changes.

1-As suggested by the referee, we have included the VMC results for J_D=0 in Fig.4 and 5.

2-Treated separately, both iPESS and iPEPS datasets locate the transition at J_D/J around 0.04-0.05. Combined together, on the basis of their mutual convergence in the large-D limit, they provide a robust estimate of the magnetization curve. We have added a comment in the revised manuscript.

3-Within the VMC approach, the dynamical structure factor is obtained by the Lehmann representation of Eq.(13), in which the sum is restricted to the discrete set of O(L^2) variational excited states defined in Eq.(9). Therefore, providing a reliable estimate for the boundaries of the continuum is not possible, because the latter is represented by a finite set of delta functions. The broad peak observed at Gamma' is located above the magnon modes and can be thus expected to lie inside the continuum of excitations. This point is discussed in the new version of the manuscript.

4-As suggested by the referee, we have improved the discussion about the differences between our results and the ones of the previous works.

---

## Round 1 · Referee Report · Samuel Bieri (Referee 2) · 2023-1-30

Strengths

1-They find accurate ground-state phase diagram of the kagome-DM quantum spin model, using advanced tensor network and variational Monte Carlo algorithms.
2-They extend calculation of dynamical spin structure factor using projected wavefunction technique to a spin-rotation broken model.
3-They provide plausible interpretations for recent inelastic neutron scattering data on three kagome compounds.

Weaknesses

1-Minor typos should be corrected and clarifications added.
2-Dynamical VMC method would profit from some further clarifications and - possibly - crosschecks.

Report

I think that the manuscript is well presented, and it gives a convincing and thorough investigation of the kagome Heiseberg model with DM interaction. It contains new and interesting results on its phase diagram, ordered moments and dynamical spin structur factor, using state-of-the art numerical techniques. I would therefore advise the manuscript to be published. Below, I have a few remarks and questions for the authors to address before publication.

Requested changes

1-First paragraph on page 2. I was a bit confused by the statement on the critical $J_D$: The formula “$J_D \lesssim 0.012(2)$” would suggest that Ref. [23] found an upper bound on the critical value. The subsequent sentence “for smaller values of $J_D$” then seems strange. Smaller than what? As far as I understand, a more precise citation of Ref. [23] would be: “$J_D\simeq 0.012(2)$”. 2-Page 9, local moment. The sublattice magnetization is calculated differently in TN and in VMC. Can the authors elaborate on the technical reasons for this? I guess, calculation of long-range correlation is not possible for TN because of small system size. Furthermore, I belive (?) that both VMC and TN explicitely break the $S_z$-rotation of the model. So, in practice, would it not be (more) meaningfull to compute the local $|\langle\vec{S}_j\rangle|$ with both methods, and compare them? Background: I wonder where the discrepancy between the two methods come from. Is it possible that TN provides a lower bound to the true magnetization? 3-I have some questions on the calculation of the transverse structure factor, and its robustness (Appendix C; pages 18-20). I am not sure if similar questions apply to the case of the out-of-plane structure factor. 3a-From the line after Eq. (23), I infer that all excited states are normalized to the same auxiliary state $|\Psi_0'\rangle$. According to the paragraph after Eq. (20), this state is the Fermionic ground state, projected to the component with $S_z=1$. Here, I wonder if this construction fails to work in case the auxilliary Hamiltonian conserves $S_z$? 3b-The MC sampling is ultimately done in the state $|\Psi_0'\rangle$. So, this state seems to enter the algorithm in a crucial way. Is there only one possible choice of $|\Psi_0'\rangle$? A priori, it seems to me that there should be other choices as well, that would similarly allow efficient VMC measurments. For example, one could probably use any state in the sum of Eq. (15). However, the final result should not depend on this choice. Have the authors checked this? I excpect checking this point thoroughly is probaly out of scope for this paper. However, the authors could still comment on it. 3c-After Eq. (24), the authors write that calculating the factor $r = ||\Psi_0'||/||\Psi_0||$ is not feasible. Maybe a more precise statement would be that it is hard to compute it directly. In fact, the authors can calculate $r$ thanks to the sum rule. As a sanity check, one should verify if $r$ it is indeed independent of momentum $q$, as claimed. Have the authors verified this? This should be easy to do. If I am not mistaken, it can be calculated from: $r^2 = S^{\pm}q/(\sum_n |S^+|^2)$. 4. There are small formatting errors in the abstract and in the discussion: "$0.03\div 0.04$" should probably be "$0.03 - 0.04$".

  • validity: top
  • significance: top
  • originality: top
  • clarity: high
  • formatting: excellent
  • grammar: excellent

Author:  Francesco Ferrari  on 2023-02-20  [id 3380]

(in reply to Report 2 by Samuel Bieri on 2023-01-30)

We thank the referee for their assessment of our work and we reply to their questions, addressing the requested changes.

1-We thank the referee for reporting this typo. We corrected it.

2-Within iPEPS, for any finite bond dimension D (in the limit of the environment dimension going to infinity) |<S>|^2 and the long-distance limit of the spin-spin correlations give equivalent estimates of m^2. On the other hand, within VMC, the wave function includes the projection operator onto the Sz=0 sector, which implies <Sx>=<Sy>=0, suitable for finite system sizes. As a result, a trustable estimator for m^2 is the long-distance limit of the spin-spin correlations, which proved to be reliable in other cases, e.g. on the square lattice [F. Ferrari, F. Becca, Phys. Rev. B 98, 100405(R) (2018)]. Concerning the discrepancy between the two methods, a similar result is observed in the ordered phase of the square lattice J1-J2 model [J. Hasik, D. Poilblanc, F. Becca, SciPost Phys. 10, 012 (2021)], but concluding that TNs provide an actual lower bound for m^2 is not possible.

3a-Yes, in the case of an auxiliary Hamiltonian conserving Sz the present scheme cannot be directly applied. Nevertheless, an alternative approach could be possible, where different fermionic orbitals are filled in the Slater determinant such that the desired value of Sz=1 is obtained.

3b-There are other possible choices for the sampling wave function, as for example the one used in [B. Dalla Piazza et al, Nat. Phys. 11(1), 62 (2015)]. At variance with the latter case, the present approach has the advantage of requiring a single Monte Carlo run for the calculations of the excited states at all desired momenta. The equivalence of the different sampling scheme has been shown, for example, for the square lattice Heisenberg model [F. Ferrari, F. Becca, Phys. Rev. B 98, 100405(R) (2018) - B. Dalla Piazza et al, Nat. Phys. 11(1), 62 (2015)].

3c-Following the suggestion of the referee, we will include the word "directly". We can indeed compute the ratio "a posteriori", enforcing the sum rule. We have performed the sanity check suggested by the referee and verified that the ratio is independent of the momentum q (within statistical errors due to the Monte Carlo sampling).

4-We have modified the notation as suggested by the referee.

---

## Round 2 · Referee Report · Samuel Bieri (Referee 2) · 2023-2-22

Report

I think that the authors have responded to all questions of the referees in a convincing way. I therefore suggest the manuscript to be published in this form.

---

## Round 2 · Referee Report · Anonymous (Referee 3) · 2023-3-3

Report

The authors have taken the referee remarks into account and provided a revised manuscript that I can now recommend for publication.

---

## Round 2 · Referee Report · Philippe Mendels (Referee 1) · 2023-3-13

Report

The authors have replied to all comments except two, see below. The first one can be discarded, the second one, I leave it at this stage as optional.

1- My item 4 of the previous report (~add a Zeeman term to the Hamiltonian) has certainly been considered by the authors as an avenue for future work. This sounds fair.

2- I cite one statement from my item 1 of the previous report "In any case, the authors would predict that the QSL state should be gapped, whereas this does not seem to be the case [P. Khuntia et al., Nature Physics 16, 469-474 (2020)], an issue still debated; maybe I missed the point but it would be useful to have that value of the gap (possibly vs J_D/J)". My comment is: Can the value of the gap be clearly plotted in Appendix B, fig. 13 ?

  • validity: -
  • significance: -
  • originality: -
  • clarity: -
  • formatting: -
  • grammar: -

Author:  Francesco Ferrari  on 2023-03-14  [id 3476]

(in reply to Report 4 by Philippe Mendels on 2023-03-13)
Category:
answer to question

Concerning the second point of the referee, we mention that a numerical evaluation of the gap size in the thermodynamic limit is an extremely demanding task, because it requires several (additional) calculations of the dynamical structure factor on different lattice sizes and for different values of J_D/J. Furthermore, within the spin-liquid region very long numerical simulations are necessary in order to obtain small statistical errors. For this reason, we would like to defer the precise numerical evaluation of the gap to a possible future work.
To further clarify this point, we will include a sentence in the appendix of the revised manuscript to remark that the presence of a gapped spin-liquid phase is inferred from the spectrum of the *unprojected* Hamiltonian, while the precise estimation of the gap can be obtained only by a numerical calculation of the Gutzwiller-projected states, which goes beyond the scope of this work.

---

## Round 2 · Author Response

Dear Editor,

we would like to resubmit the revised version of the manuscript which includes changes addressing the requests made by the referees. We have replied to the comments of the referees and highlighted the changes made to the manuscript.

Sincerely yours,
Francesco Ferrari (on behalf of all the authors)

---

## Round 2 · List of Changes

-we included the revised estimates of J_D/J for Herbertsmithite (Section 1)
-we updated Fig.4 and 5 to include the results for J_D=0
-we improved the discussion of the results for the magnetization in comparison to previous works based on exact diagonalization (Section 3.1)
-we commented about the possible separate fitting of iPEPS and iPESS data (Section 3.1)
-we improved the discussion about the broad peak observed in our spectra at Gamma' above the magnon modes and the comparison to experimental results (Section 3.2)
-we improved the discussion in the appendices (mostly Appendix A); we updated Fig.11-12, correcting minor typos and improving the clarity of the plots
-we added a paragraph to discuss the comparison of our spectra to previous results at the end of Appendix C
-we corrected minor typos throughout the manuscript
-we added new references

---

## Editorial Decision

published